# Vitamin D Effects on Bone Homeostasis and Cardiovascular System in Patients with Chronic Kidney Disease and Renal Transplant Recipients

**DOI:** 10.3390/nu13051453

**Published:** 2021-04-25

**Authors:** Giuseppe Cianciolo, Maria Cappuccilli, Francesco Tondolo, Lorenzo Gasperoni, Fulvia Zappulo, Simona Barbuto, Francesca Iacovella, Diletta Conte, Irene Capelli, Gaetano La Manna

**Affiliations:** Nephrology, Dialysis and Renal Transplantation Unit, IRCCS Azienda Ospedaliero-Universitaria di Bologna, 40138 Bologna, Italy; giuseppe.cianciolo@aosp.bo.it (G.C.); maria.cappuccilli@unibo.it (M.C.); francesco.tondolo@studio.unibo.it (F.T.); lorenzo.gasperoni3@gmail.com (L.G.); fulvia.zappulo@studio.unibo.it (F.Z.); simona.barbuto@studio.unibo.it (S.B.); francesca.iacovella3@unibo.it (F.I.); diletta.conte3@unibo.it (D.C.); irene.capelli4@unibo.it (I.C.)

**Keywords:** cardiovascular risk, chronic kidney disease, CKD–MBD, dialysis, kidney transplant, vitamin D

## Abstract

Poor vitamin D status is common in patients with impaired renal function and represents one main component of the complex scenario of chronic kidney disease–mineral and bone disorder (CKD–MBD). Therapeutic and dietary efforts to limit the consequences of uremia-associated vitamin D deficiency are a current hot topic for researchers and clinicians in the nephrology area. Evidence indicates that the low levels of vitamin D in patients with CKD stage above 4 (GFR < 15 mL/min) have a multifactorial origin, mainly related to uremic malnutrition, namely impaired gastrointestinal absorption, dietary restrictions (low-protein and low-phosphate diets), and proteinuria. This condition is further worsened by the compromised response of CKD patients to high-dose cholecalciferol supplementation due to the defective activation of renal hydroxylation of vitamin D. Currently, the literature lacks large and interventional studies on the so-called non-calcemic activities of vitamin D and, above all, the modulation of renal and cardiovascular functions and immune response. Here, we review the current state of the art of the benefits of supplementation with native vitamin D in various clinical settings of nephrological interest: CKD, dialysis, and renal transplant, with a special focus on the effects on bone homeostasis and cardiovascular outcomes.

## 1. Introduction

Vitamin D comprises a group of liposoluble secosterols, among which the two main forms are vitamin D2 (ergocalciferol) and vitamin D3 (cholecalciferol); the first is derived from the diet while the latter is synthesized in the skin through UV irradiation of 7-dehydrocholesterol to provitamin D3, with a further thermal isomerization step to form vitamin D3 [1]. The main steps necessary for vitamin D activation are 25-hydroxylation, 1α-hydroxylation, and 24-hydroxylation, all of them catalyzed from oxidases (CYPs) of cytochrome P450. The native vitamins D2 and D3 are carried bound to vitamin D binding protein (DBP) into the liver, where hydroxylation at the C25 position occurs, thus forming 25(OH)D (25-hydroxyvitamin D), the principal circulating form of vitamin D. However, the levels of circulating levels of 25(OH)D are only partly dependent on the liver because different other tissues, like the skin and testes, express this specific enzymatic activity [2,3].

At the renal level, 25(OH)D is filtered by the glomerulus and then actively reabsorbed into renal tubular cells by DPB receptors (megalin and cubilin), and then converted into 1,25-(OH)_2_D (1,25-dihydroxyvitamin D, also known as 1,25-dihydroxycholecacliferol or calcitriol) by the enzyme 1α-hydroxylase (CYP27B1). CYP27B1 activity in the kidney is stimulated by parathyroid hormone (PTH) and suppressed by fibroblast growth factor 23 (FGF23) and 1,25(OH)_2_D itself. Moreover, hyperphosphatemia, both directly and via FGF23, can also suppress activity CYP27B1 at the renal level [4,5]. 

The role of vitamin D as a hormone has been classically related (at least until recently) to bone physiology and mineral homeostasis. Vitamin D participates with other three hormones, PTH, FGF23, and calcitonin, in the regulation of calcium and phosphate metabolism in different target organs, like bone, kidneys, liver, and gastrointestinal tract. Therefore, its deficiency often results in bone mineral disorders and in the development of secondary hyperparathyroidism (SHPT) due to the decreased parathyroid levels of vitamin D receptor (VDR) and calcium sensing receptors (CaSRs) with consequent reduction of the inhibitory stimuli on PTH secretion and parathyroid sensitivity to ionized calcium [6,7,8]. 

However, other extrarenal cells, like osteoblasts, osteoclasts, and parathyroid cells, can also synthesize calcitriol, due to their ability to express megalin, cubilin, and CYP27B1 [9,10,11].

It is important to note that the 1,25(OH)_2_D produced locally by non-renal tissues is involved in autocrine/paracrine rather than endocrine functions and does not modify the circulating levels of calcitriol [12]. Indeed, while renal cells provide for the synthesis of calcitriol necessary to regulate the hormonal activity involved in the regulation of mineral metabolism, the other cells of the body can provide for the synthesis and metabolism of calcitriol necessary for the functioning of the cell itself. Thus, 1,25(OH)_2_D is regarded as the active form for its ability to bind VDR [13].

VDR is nearly ubiquitously expressed and most of the cells are sensitive to 1,25(OH) 2D, with a different degree of activity among them [14,15].

Due to this condition, it is possible to speculate on different mechanisms of action of vitamin D other than the usual action on bone. At present, the literature lacks large and interventional studies on the so-called non-calcemic activities of vitamin D, in particular the regulation of renal and cardiovascular functions and immune response. Moreover, different recent studies do not agree on the action of vitamin D supplementation on PTH suppression.

The aim of this review is to investigate bone effects of vitamin D supplementation in various clinical settings of nephrological interest: early CKD stages, dialysis, and renal transplant. We also analyzed the effects of vitamin D supplementation and vitamin D receptor agonist (VDRA) therapy in extra-bone settings, like proteinuria and cardiovascular disease. This work also revised the controversial key elements regarding vitamin D therapy applications in clinical practice.

## 2. Role of Vitamin D on Mineral and Bone Homeostasis in Health and CKD

Bone remodeling is a continuous process where mature bone tissue is removed from the skeleton (a process called bone resorption) and new bone tissue is formed (a process called new bone formation). These processes also control the reshaping or replacement of bone following injuries like fractures, but also microdamage, which occurs during normal activity [16]. Bone remodeling is tightly regulated by local and systemic hormones, such as PTH, 1,25(OH)_2_D, Wnt signaling pathways, FGF23, and vitamin K (VK). Under the effects of these hormones, bone remodeling is ultimately the expression of the activity and the interplay of two cellular populations: osteoblasts that control the bone formation and osteoclasts, with the ability to resorb mineralized bone [17]. In this complex scenario, vitamin D plays a pivotal role.

The main endocrine function that follows the activation of VDR is the regulation of mineral and bone homeostasis in intestinal, renal, and bone tissues. VDR activation controls the calcium and phosphate absorption at the intestinal level, the calcium tubular reabsorption at the renal level, and the activity and viability of bone cells [18]. These effects follow a wide range of biological actions mediated by vitamin D responsive elements (VDREs) and lead to changes in the expression of many genes, e.g., receptor activator for nuclear factor κ b ligand (RANKL), low-density lipoprotein receptor-related protein 5 (LRP5), cytochrome P450 family 24 subfamily A member 1 (CYP24A1), and transient receptor potential cation channel subfamily V member 6 (TRPV6) [19].

Indeed, in bone cells, 1,25(OH)_2_D inhibits Bmp2 gene expression and upregulates FGF23 [20]; the latter, at the same time, represses 1α-hydroxylation in kidneys targeted to prevent a continuous activation of FGF23 by 1,25(OH)_2_D. FGF23 causes, in turn, the inactivation of the Wnt signaling pathway in osteoblasts thanks to the upregulation of dickkopf-related protein 1 (Dkk1) via the MAPK pathway. Dkk1 acts both in an autocrine and paracrine way, promoting the phosphorylation of β-catenin responsible for the inhibition of the Wnt pathway [21].

As demonstrated by Pereira et al., 1,25(OH)_2_D exerts a different effect on osteoblasts from healthy controls as compared to CKD patients. Moreover, 1,25(OH)_2_D promotes osteoblast maturation, increasing the expression of osteocalcin, a marker of maturation in both healthy controls and CKD osteoblasts. In CKD osteoblasts, 1,25(OH)_2_D also stimulates the expression of the osteoclast differentiation factor RANKL and, albeit in high doses, increases the RANKL/osteoprotegerin ratio. Moreover, in the bone of CKD patients, 1,25(OH)_2_D has a negligible effect on the expression of the early osteoblast marker, Runt-related transcription factor 2 (RUNX2), and alkaline phosphatase. These data suggest that 1,25(OH)_2_D may play an important role in osteoblast maturation by regulating osteoclast–osteoblast coupling in the bone of CKD patients [22].

Calcitriol also influences the activity of the canonical Wnt (or Wnt/β-catenin) pathway by increasing the expression of LRP5, a gene that stimulates osteoblast proliferation via enhanced canonical Wnt signaling, and therefore has an anabolic effect on the bone [23]. Notwithstanding, calcitriol, acting in bone cells, is also known to directly upregulate gene expression of sclerostin that, in turn, carries out a negative feedback on 1,25(OH)_2_D synthesis through the downregulation of renal 1α-hydroxylase in renal and bone cells [24,25].

The effects of 1,25(OH)_2_D, FGF23 and sclerostin are illustrated in Figure 1.

Vitamin D deficiency is common in patients with CKD, whatever the stage of the disease. It has been proposed that the progressive loss of renal function results in a reduced kidney ability to synthesize active vitamin D and to eliminate phosphorus. Other reasons for the poor vitamin D status in uremic patients are the reduced nutritional intake associated with dietary restrictions (e.g., low-protein and low-phosphate diets), lack of appetite, and gastrointestinal symptoms, together with inadequate sun exposure linked to decreased mobility [26,27]. 

In CKD patients, the dysregulated mineral homeostasis has not only an impact on the skeletal system but is also strictly associated with other important events: the development of vascular calcification and, above all, cardiovascular disease. As specified below (see Section 4.3. “Effects of VDRAs on LVH in CKD and dialysis”), these consequences are related to the other effects than bone effects of both native vitamin D and calcitriol; in fact, the latter exerts several other functions, like inhibition of angiogenesis, stem cell differentiation, apoptosis, renin production, and induction of insulin release [28,29,30]. 

Recently, evidence has emerged about the synergistic interplay between vitamins D and K on bone and cardiovascular health. Vitamins D and K share osteo-inductive properties and, in particular, some studies demonstrated that menaquinones (MKn) enhance vitamin D3-induced mineralization by increasing bone Gla protein (BGP) gene expression, as well as BGP protein content in the extracellular matrix. BGP has an essential role in the synthesis and regulation of bone matrix, in addition to the control bone mineralization. BGP can also carry out a mechanical function within the bone matrix since it binds hydroxyapatite and forms a complex with collagen, acting as a bridge between the matrix and mineral component of bone tissue [31,32]. Furthermore, BGP protects against vascular calcifications; decreased BGP levels have been found in hemodialysis patients with one or more vertebral fractures and vascular calcifications and have been associated with accelerated abdominal aortic calcification and higher mortality in the elderly [33]. 

Calcitriol also increases matrix Gla protein (MGP) mRNA and raises MGP secretion up to 15 times. The MGP gene promoter contains a vitamin D response element, able to enhance MGP expression two- to three-fold after vitamin D binding [34]. MGP expression is also induced significantly by RANKL, whose synthesis in osteoblastic cells is stimulated by 1,25(OH)_2_D through a vitamin D-responsive element. In addition to being considered one of the most effective endogenous inhibitors of vascular calcification, MGP plays a multifaceted role in bone turnover, since it regulates not only bone formation but also osteoclast differentiation and bone resorption [35].

## 3. Controversial Key Elements Regarding Vitamin D Status Assessment and Therapy Applications in Clinical Practice

In the general population, as well as in CKD patients, the measurement of circulating 25(OH)D is considered the most reliable indicator of vitamin D status due to the long half-life and its direct connection with UV-induced cutaneous production and exogenous dietary intake [36]. Since vitamin D status shows seasonal variations related to the extent of sun exposure, circulating 25(OH)D levels should be measured during the winter and the summer months [37,38]. 

Concerning the general population, the Endocrine Society Guidelines indicate 25(OH)D concentrations above 30 ng/mL as normal, concentrations between 21 and 29 ng/mL as vitamin D insufficiency, and levels <20 ng/mL as vitamin D deficiency [39]. 

It is well established that PTH levels are inversely correlated with circulating 25(OH)D, but in the uremic state, they can be affected by several factors, like calcium intake, fasting status, seasonal changes, race, age, pregnancy, and drugs (i.e., thiazide diuretics) [40,41,42]. 

Over the years, many international guidelines have suggested different target levels of 25(OH)D to define vitamin D deficiency and insufficiency in the CKD population [43]. 

In 2003, the Kidney Disease Outcomes Quality Initiative (KDOQI) international guidelines indicated target vitamin D levels ≥ 30 ng/mL to prevent SHPT-related mortality and to reduce fracture risk in patients with CKD stages 3 and 4 [44]. 

In 2016, a position paper on vitamin D in CKD patients of the working group “Trace Elements and Mineral Metabolism” of the Italian Society of Nephrology stated that, regarding the current debate on 25(OH)D optimal levels, achieving concentrations above 30 ng/mL appears to be adequate to avoid vitamin D deficiency and treat SHPT, both in non-dialysis-dependent CKD patients and in dialysis-dependent patients [45]. 

More recently, a report on a “controversies conference” on vitamin D in CKD, promoted by the National Kidney Foundation, established that nephrologists should classify 25(OH)D levels >20 ng/mL as “adequate”, regardless of PHT value, while 25(OH)D concentrations <15 ng/mL should be treated. Patients with 25(OH)D levels ranging from 15 to 20 ng/mL might receive vitamin D supplementation or not, based on counterregulatory PTH activity. The conference also underlined how nutritional vitamin D (ergocalciferol, cholecalciferol, or calcifediol) should be given prior to supplementation with active vitamin D compounds (vitamin D receptor activators, VDRAs) [46]. 

Besides the evaluation of vitamin D status assessment, vitamin supplementation in CKD patients, both non-dialysis dependent and dialysis dependent, as well as in KTRs, is still a debated point. Firstly, even if higher doses lead to better vitamin D status, no clear data are available about the cumulative recommended daily or weekly intake. Secondly, nutritional vitamin D supplementation is useful to prevent SHPT, but not to reduce persistently increased PTH levels. Thirdly, another issue to be fully elucidated is that of the timing and safety of VDRA therapy. Lastly, there are contrasting views about the benefits underlying vitamin D administration in terms of kidney transplant outcome: reaching vitamin D adequacy levels or regression of PTBD?

To clarify the discrepancies between the several trials testing the effects of nutritional vitamin D supplementation (ergocalciferol or cholecalciferol) in CKD, dialysis, and KTRs, Kandula et al. analyzed vitamin D supplementation in a large meta-analysis involving non-dialysis-dependent patients, dialysis-dependent patients, and transplant recipients. The results showed a more pronounced PTH suppression in dialysis patients compared to KTRs [47]. On the other hand, further studies demonstrated no reduction in PTH levels in patients receiving vitamin D supplementation compared to placebo. 

The next paragraphs aim at responding, as far as possible, to these observations.

### 3.1. Vitamin D in Non-Dialysis-Dependent CKD Patients: The Effect on Vitamin D Status and SHPT

The KDIGO guidelines suggest nutritional vitamin D supplementation to improve poor vitamin D status and to prevent SHPT in CKD patients, although the currently available data are still inconclusive, due to the mainly observational nature of the studies.

As detailed in Table 1, in recent years, several authors have retrospectively analyzed the effects of supplementation with native vitamin D in CKD patients, with conflicting results [48,49,50,51]. In all these studies, none of the treatments led to serum calcium and phosphate variations or adverse effects. Instead, discordant data were found in terms of improvement of vitamin D status and control of SHPT. It appears that higher doses of vitamin D supplementation result in a better vitamin D status. However, the link between vitamin D therapy and SHPT is not actually clear. 

Vitamin D supplementation represents a relevant point of discussion for various international guidelines: in the Canadian healthcare system, the need for vitamin D supplementation is linked to CKD stages; as matter of fact, vitamin D supplemention of 400 IU is recommended in CKD G3 patients, while no recommendations are present for CKD G4-G5 patients [52]. 

The effects of vitamin D supplementation on vitamin D status and SHPT are related to an unclear mechanism and depend on the precocity of the treatment. There is discordant evidence about the 25(OH)D levels able to induce a reduction of PTH levels in CKD patients [53]. It is important to underline that vitamin D supplementation is effective in preventing an increase in PTH rather than reducing high levels of PTH in advanced-stage CKD patients [53,54,55]. 

If it is true that parathyroid glands express 1α-hydroxylase, a possible autocrine mechanism by which vitamin D supplementation reduces PTH production should also be considered. At the same time, data from in vitro studies indicate that 25(OH)D can directly suppress PTH secretion via VDR activation [55,56]. 

The growing interest in a novel modified release formulation of calcifediol is due to its safer pharmacokinetics and better bioavailability. 

Sprague et al. investigated the impact of extended release calcifediol (ERC), a formulation orally administered as a prohormone of active vitamin D, in 429 patients with stage 3–4 CKD, poor vitamin D status, and SHPT. The authors showed that after 26 weeks, ERC therapy resulted in a progressively better pattern of circulating 1,25(OH)_2_D levels, plasma PTH, and serum bone turnover markers, independent of CKD stage and without adverse repercussions on safety parameters. However, these positive results were observed only after the achievement of 25(OH)D levels above 50.8 ng/mL, suggesting that the targets used for vitamin D repletion therapy in CKD may be inadequate [54]. 

The currently available clinical data regarding the pharmacokinetics, pharmacodynamics, efficacy, and safety of ERC in CKD–MBD were recently reviewed in non-dialysis-dependent patients [57] and in those with different CKD stages, including dialysis-dependent patients [58]. 

This new modified release formulation of calcifediol represents a safe and promising option for the management of SHPT due to its efficacy in gradually increasing both 25(OH)D and 1,25(OH)_2_D levels, avoiding CYP24A1-mediated vitamin D inactivation, and in PTH suppression, with no or minimal negative rebounds of serum phosphorus, calcium, and FGF23 [57,58]. 

Patients with impaired renal function have reduced activity of 1α-hydroxylase (CYP27B1), therefore, nephrologists have been traditionally more prone to treat CKD patients with hypovitaminosis D using activated vitamin D or related analogs, including selective VDRA, paricalcitol, or non-selective VDRA such as calcitriol, alfacalcidol, doxercalciferol, 22-oxacalcitriol, and maxacalcitol [59]. 

According to KDIGO guidelines, the use of calcitriol and vitamin D analogs should be reserved for patients with severe and progressive hyperparathyroidism [44]. 

The position statement of the Italian Society of Nephrology suggests starting active vitamin D therapy in patients with CKD stages above 3 and high-serum PTH and normal 25(OH)D levels, in the absence of hypercalcemia and/or hyperphosphatemia [45]. 

Palmer et al. published a systematic review to investigate the role of active vitamin D compounds on PTH suppression in non-dialysis CKD patients (16 studies with 894 patients were selected). All the examined vitamin D formulations adequately reduced serum PTH, without affecting mortality risk or need for dialysis, regardless of administration route or therapy schedule [60]. In CKD patients, the importance of active vitamin D is also related to the action on vascular calcifications, as its protective or inducing effects depend on the dose of vitamin D. Indeed, experimental and clinical studies indicated that more physiological doses vitamin D can be safely administered, while elevated doses might lead to a worsening of vascular calcifications [61].

As reported below, the action of vitamin D supplementation is even more important if we consider the role in the endocrine system and metabolic bone disease; the ubiquity of vitamin D receptors is one of the reasons why the role of vitamin D cannot be reduced to its action on bone metabolism [62].

### 3.2. Vitamin D in Dialysis-Dependent Patients: The Effect on Vitamin D Status and SHPT

Data regarding the effects of 25(OH)D on hard outcomes, on markers of mineral metabolism, and on SHPT control in dialysis patients are still inconsistent. Moreover, the variability in clinical practice and the lack of prospective studies examining the safety and efficacy of nutritional vitamin D replacement in ESRD further complicate the scenario [63]. Therefore, despite the high prevalence of 25(OH)D deficiency in dialysis patients [13], controversies still exist about dosage and timing for nutritional vitamin D administration in this population [13,63].

A recent large meta-analysis including four RCTs of 130 non-dialysis CKD patients and 14 RCTs including 888 participants on maintenance dialysis did not find significant differences in PTH reduction between nutritional vitamin D (cholecalciferol) and placebo [64].

Successive RCTs with a significant number of hemodialysis patients failed to demonstrate a PTH-lowering effect by ergocalciferol or cholecalciferol compared to placebo even if sufficient 25(OH)D levels were reached [63,65]. The DIVINE trial randomized 105 hemodialysis patients from 32 US centers with 25(OH)D levels ≤ 32 ng/mL, to high or low ergocalciferol dose or placebo. At the end of the 12-week treatment period, the primary endpoint of reaching 25(OH)D sufficiency was achieved, but no effects on SHPT, hospitalizations, and all-cause mortality were observed [63].

Different results were found in a one-year prospective study including 158 hemodialysis patients that compared oral cholecalciferol administered once or thrice weekly for 6 months, after each dialysis session, with a dosage based on patients’ 25(OH)D levels. Cholecalciferol therapy resulted in a better serum 25(OH)D and 1,25(OH)_2_D status, together with increased albumin levels, while a significant reduction was observed for calcium, phosphorus, intact PTH, C-reactive protein, brain natriuretic peptide, and left ventricular mass index at the end of the supplementation period [66]. 

Given the uncertain therapeutic index and the concerns regarding the appearance of hypercalcemia and hyperphosphatemia following the use of VDRAs, newer vitamin D compounds have been evaluated over the years. 

The safety of calcidiol supplementation in hemodialysis patients already under VDRA therapy was recently analyzed by Villa-Bellosta et al. in a two-year observational cohort study on 129 patients. The results showed that the patients under therapy with calcitriol alone or combined with paricalcitol were associated with significantly higher mortality rates than the untreated ones, suggesting caution in calcidiol/paricalcitol treatment in hemodialysis patients [67]. 

A systematic review by Palmer et al. analyzed the results from RCTs (a total of 2773 hemodialysis patients) on the effects of calcitriol, alfacalcidol, 24,25(OH)_2_ vitamin D3, doxercalciferol, maxacalcitol, paricalcitol, and falecalcitriol on SHPT control. Despite heterogeneity in the outcomes, active vitamin D compounds suppressed serum PTH, but at the expense of a rise in serum phosphorus and calcium (although the latter was not statistically significant). Moreover, novel vitamin D compounds (paricalcitol, maxacalcitol, doxercalciferol) did not prove to be safer in terms of calcemic status compared to calcitriol or alfacalcidol. However, all the formulations, administration routes, and therapeutic schemes had no clear positive implications in terms of mortality rate, bone pain, or need for parathyroidectomy [68].

### 3.3. Vitamin D in Kidney Transplant Recipients: The Problem of Post-Transplantation Bone Disease and the Effect on Vitamin D Status

CKD–MBD is still present after kidney transplantation and it is dependent both on previous bone damage and de novo risk factors. One of the most important bone metabolic alterations is represented by hyperparathyroidism; this condition could be a maladaptive response (persistent hyper-PTH) or a compensatory response (de novo hyper-PTH) [69]. 

There is a large body of evidence from experimental and clinical studies to support the association between reduced 25(OH)D levels and deranged vitamin D metabolism in kidney transplant recipients (KTRs). It has been observed that only 12% of KTRs show normal 25(OH)D levels (>30 ng/mL), while deficiency and insufficiency are found with a rate ranging between 30% and 81%, respectively [70]. 

CKD–MBD after kidney transplantation is widely present and it is characterized by different metabolic and bone conditions, named post-transplantation bone disease (PTBD). PTBD results in poor bone quality, higher fracture risk, accelerated vascular aging, and deterioration of allograft function [71,72]. PTBD has a multifactorial etiopathogenesis which involves pre-existent CKD–MBD, derangements in the PTH–FGF23–vitamin D axis, metabolic alterations in calcium/phosphorus balance, and immunosuppressive therapy [69]. Commonly, at one year after kidney transplantation, high PTH levels are present in 30–60% of recipients [73]. 

Among immunosuppressive therapies, steroids are an important cause of bone alterations by different pathways. The mechanism by which corticosteroids alter vitamin D metabolism are not completely understood; steroids express enzymes involved in vitamin D catabolism and increase PTH and FGF23 levels. In this context, the cumulative steroid dose is important, and it is inversely related with 1,25(OH)_2_D levels; steroids reduce 1,25(OH)_2_D levels in an indirect mechanism too, inducing high FGF23 levels [71]. 

Besides steroids, other immunosuppressive agents have also been implicated in the dysregulation of vitamin D metabolism seen in KTRs, in a drug class-dependent rather than non-drug concentration-dependent way. There is much evidence to indicate that calcineurin inhibitor (CNI) therapy is inversely associated with lower 25(OH)D levels, while mTOR inhibitors do not seem to influence vitamin D status [74,75]. 

A possible explanation might lie in the suppression of the 25-hydroxylase activity of liver CYP3A4 by CNIs, resulting in lower levels of circulating 25(OH)D; but, at the same time, there is a possible role of the downregulation of VDR induced by CNIs [76]. 

Therefore, KTRs are burdened by a combination of alterations in bone remodeling, including impaired bone mineralization and osteoblast function, expressed as reduced osteoblastogenesis, increased osteoblast apoptosis, and enhanced bone resorption, leading to bone loss [70,77].

There are poor and controversial data and no randomized controlled trials regarding the role of nutritional vitamin D therapy in PTBD and in renal transplant outcomes. Firstly, for vitamin D assessment, it is important to emphasize that 25(OH)D levels in KTRs should be interpreted in the same way as in CKD and the general population by clinicians [78]. As for the safety and effectiveness of nutritional vitamin D in KTRs, it appears clear that its supplementation is the first step in PTH suppression, but no data are available on the effects on important endpoints, like cardiovascular disease, death, and PTDB prevention. The most important studies are reported in Table 1. 

Recent evidence has revealed the efficacy of a monthly supplementation of 25,000 IU cholecalciferol in lowering PTH levels, but not in preventing PTBD [79]. Conversely, other reports indicate that a daily combination of 400 IU/day oral cholecalciferol and 600 mg/day oral calcium can normalize serum PTH levels and reduce lumbar spine, femoral neck, and femoral total bone loss after a 12-month post-transplant treatment [85]. According to other studies, cholecalciferol appears to be effective in ameliorating SHPT and renal transplant outcomes, but the normalization of vitamin D status to a concentration >30 ng/mL requires high-dose treatment [80].

As for the dosage problem of vitamin D supplementation, according to the Canadian healthcare system, 400 IU daily of nutritional vitamin D is recommended in KTRs with CKD G1–G3 [52].

As for CKD patients, ERC was tested in KTRs too, with optimal results depending on the different administration schedules [86]. 

Active vitamin D compounds may ameliorate post-transplantation CKD–MBD by many mechanisms, feasibly suppressing PTH secretion, reducing glucocorticoid-induced decreases in intestinal calcium absorption, and promoting differentiation of osteoblast precursors into mature cells. There is proof to indicate that VDRAs can decrease PTH levels and improve CKD–MBD after kidney transplant, but most of the studies do not consider other clinically critical issues, like mortality, hospitalizations, or fractures [87].

Therapy with low-dose calcium supplements during 1 year, combined with intermittent calcitriol for the first 3-months period after renal transplant can be safety administered in KTRs, achieving bone loss prevention at the proximal femur [81]. 

## 4. Extra-Bone Effects of Vitamin D Supplementation

The action of both nutritional vitamin D and VDRA is not limited on bone, but it is extended to systemic functions.

### 4.1. Nutritional Vitamin D and VDRA Effects on Proteinuria

One of the main extra-bone effects of vitamin D is represented by the antiproteinuric effect of native vitamin D and VDRA mediated firstly by dysregulation of the renin–angiotensin–aldosterone system. 

The inverse correlation between calcitriol and renin is a consolidated idea in the literature. Forman et al. demonstrated that patients with low 25(OH)D levels (<30 ng/mL) had higher renin and angiotensin II levels because of an inadequate activation of the renin angiotensin system (RAS). It seems that this inverse regulation is independent of calcium and PTH levels [88]. 

Since mineralocorticoid receptors belong to the same superfamily of VDR, calcitriol binds to VDR, in both an agonistic and synergic way, thus reducing renin expression. 

The data of several studies report different levels of support for the use of vitamin D in an antiproteinuric way, in association or in substitution of anti-RAS agents [78]. Both native vitamin D and active vitamin D are considered as a safe antiproteinuric agent in CKD patients and KTRs. 

A prospective crossover RCT compared the efficacy of a 6-month treatment with paricalcitol (1 µg/day for 3 months and then 2 µg/day if tolerated) or non-paricalcitol therapy on PTH levels (primary endpoint), mineral metabolism, and proteinuria in 43 renal transplant recipients with SHPT. After this treatment period, serum PTH, bone-specific alkaline phosphatase and osteocalcin, urinary deoxypyridinoline-to-creatinine ratio, and 24 h proteinuria levels decreased only in the paricalcitol arm. Vertebral mineral bone density, evaluated by dual-energy X-ray absorption, also improved with paricalcitol therapy. Taken together, these data seem to indicate that 6-month paricalcitol supplementation ameliorates SHPT, reduces proteinuria, and attenuates bone loss and remodeling [83]. 

Recently, Pihlstrøm et al. investigated the efficacy of the early introduction of paricalcitol 2 μg/day in de novo renal transplant recipients to reduce proteinuria and prevent progressive allograft fibrosis. At the end of the 44-week follow-up period, paricalcitol treatment lowered PTH levels, without significantly reducing albuminuria or improving vascular parameters (reactive hyperemia index and pulse wave velocity), GFR, or expression profiles of genes related to allograft function [84]. 

De Sévaux et al. examined the effects of treatment with daily calcium (1000 mg) and active vitamin D (0.25 µg) on bone mineral density in 111 renal transplant recipients during the first 6 post-transplant months. The results showed that bone loss extent was significantly smaller in the treated patients, suggesting that low-dose active vitamin D and calcium can partly prevent bone loss in the early post-transplant period [82].

### 4.2. Vitamin D and Left Ventricular Hypertrophy

In past decades, large observational studies in the general population provided solid evidence about the link between deficits of vitamin D and increased risk of cardiovascular events, such as myocardial infarction [89,90,91], heart failure [92], arrhythmias [93], stroke [91,94], and cardiovascular death [95]. In the CKD population, the association of vitamin D deficiency with increased mortality and adverse cardiovascular outcome has been demonstrated in observational studies [91,96]. In this scenario, left ventricular hypertrophy (LVH) represents a powerful link [97]. LVH is the most common cardiac abnormality in CKD patients, with prevalence rates up to 75% [98]. Identified causal factors of LVH in this population are hypertension, high body mass index, and mineral metabolism impairment, including hyperphosphatemia, secondary hyperparathyroidism, vitamin D deficiency, FGF23–Klotho axis derangement [97], anemia [99], and fluid overload [100,101]. 

The main key pathophysiological mechanisms on the basis of the causal relationship between vitamin D deficiency and LVH are: (i) increased renin–angiotensin–aldosterone system activity; (ii) interaction between VDR, expressed in cardiomyocytes, and (iii) FGF23/FGFR4 signaling pathway.

#### 4.2.1. RAS Overactivation

LVH is a well-established consequence of RAS overactivation [97]. Deficient VDR activation has been linked, in experimental models, to higher expression of renin in kidneys, elevated angiotensin II and aldosterone levels, and LVH [102,103,104,105]. The reversibility of these effects after calcitriol administration confirms the experimental evidence showing that activated VDR is a powerful inhibitor of renin gene transcription through a cAMP-dependent response element in the renin gene [102,106,107]. Moreover, recent evidence supports the existence of a myocardial RAS, regulated independently of the circulating RAS, associated with LVH [108,109]. Cardiomyocytes express all of the RAS (renin, angiotensin, angiotensin receptor) and, recently, Freundlich et al. demonstrated an increased mRNA expression of renin, angiotensinogen, and angiotensin receptor 1 (AT1R) in cardiomyocytes of uremic rats [110]. Besides systemic RAS control, LVH protection, in a uremic model, was demonstrated to be mediated even by cardiac RAS gene downregulation after paricalcitol administration, reinforcing the association between myocardial RAS overactivity and LVH in renal failure [107,108,109,110]. 

#### 4.2.2. FGF23/FGFR4 and VDR

FGF23 can bind to four receptors (FGFR1–4) that are localized in different target organs [111,112,113]. Klotho is known to acts as an FGF23 co-receptor that promotes efficient binding of FGF23 to FGFRs, although its presence is not essential for FGF23 cellular activity in organs like the liver and heart [114]. Even if the role of FGF23 in vascular calcifications is not completely clear, its action on LVH is well defined [115]. In experimental models, FGF23 directly induces LVH through an FGFR signaling pathway independent of Klotho, which is not expressed in cardiomyocytes [116]. While in tissues where the co-receptor Klotho is expressed, FGF23/FGFR binding activates the Ras/mitogen-activated protein kinase (MAPK) cascade [114], and hypertrophy of cardiomyocytes is induced by the interaction between FGF23 and FGFR4 which stimulates a different transduction signal cascade involving phospholipase Cy (PLCy), calcineurin, and nuclear factor of activated T cells (NFAT). The interaction between active vitamin D and FGF23/FGFR4 is reported in Figure 2.

Evidence of such a mechanism has been provided both in experimental models with and without renal failure [116,117,118] and in CKD patients [119]. Specifically, an autoptic study in CKD patients revealed an enhanced expression of FGFR4, calcineurin, PLCy, and NFAT in the myocardium and showed a correlation of such parameters with LVH [119].

The role of calcitriol in the heart is mediated by its binding to VDR expressed in cardiomyocytes that leads to modifications of gene expression to protect from hypertrophy [120]. For instance, specific VDR deletion in cardiomyocytes in a non-CKD murine model determined LVH in the absence of hypertension or other risk factors, revealing a direct causality between VDR signaling and LVH [121]. Under physiological conditions, in cardiomyocytes, activated VDR interacts with the FGFR4 signaling cascade, at a genomic level, to maintain integrity of the heart structure. Although the exact mechanisms are not fully understood, VDR and NFAT are both transcription factors with opposite functions in cardiomyocytes and share the same promoter genomic region [122,123]. Thus, it has been hypothesized that the calcitriol/VDR interaction could determine the inhibition of pro-hypertrophic genes that would be enhanced by FGFR4/NFAT signaling [124]. The direct interaction between these pathways was demonstrated initially in T cell models where binding of NFAT with NFAT response elements was inhibited by activation of VDR by calcitriol [125]. Such experimental evidence has even recently been provided in cardiomyocytes [124,126]. In addition, calcitriol also inhibits FGFR4 signaling in a non-genomic manner by direct interaction with PLCy, determining a reduction of its activity [124]. 

Thus, under uremic conditions, two major drawbacks coexist: (a) calcitriol deficiency due to reduced kidney 1α-hydroxylase activity and to increased FGF23 [127]; (b) elevated FGF23 levels and enhanced expression of FGFR4, PLCy, calcineurin, and NFAT [124]. Therefore, in advanced CKD and dialysis, the regulatory role of calcitriol/VDR in LVH appears to be lost, allowing unfavorable cardiac remodeling.

### 4.3. Effects of VDRAs on LVH in CKD and Dialysis

Although the causal relationship between vitamin D deficit and LVH is established, the clinical effects of treatment with VDRAs on LVH in CKD and dialysis subjects are less clear and clinical trials have given unsatisfactory results. 

In kidney failure rat models, VDRAs provided significant protection from LVH development [110,124,128,129]. Leifheit-Nestler et al. demonstrated in 5/6 sub-nephrectomized rats that calcitriol protects against LVH development through inhibition of the FGFR4/PLCy/calcineurin/NFAT pathway and reduction of FGFR4 expression on cardiomyocytes. In addition, the authors showed two interesting points. First, calcitriol-induced decreased expression of cardiac FGF23 (e.g., produced by cardiomyocytes) determined an inhibition of autocrine and paracrine activation of FGFR4. Secondly, despite the increased circulating FGF23 levels released by osteocytes after calcitriol treatment, the negative interaction between VDR and FGFR4 signaling ultimately determined LVH protection [130]. Nonetheless, it must be pointed out that such effects were confirmed only for a high-dose calcitriol group and that these rats developed hypercalcemia and hyperphosphatemia as side effects [124]. 

More recently, starting from the issue that extremely high circulating FGF23 levels may prevail against VDR protective effects on LVH, Czaya et al. tested a combined treatment of paricalcitol and a pan-FGF23 receptor blocker in 5/6 nephrectomized rats. The results confirmed that paricalcitol suppresses upregulated myocardial calcineurin/NFAT target genes and showed that such effects were amplified by co-administration of the FGF23R blocker, PD173074, as elegantly underlined by a strong inverse relation between cardiac hypertrophy and the ratio of paricalcitol dose/FGF23 levels. An unexpected finding was the inhibition of the cardiac RAS by an FGFR23R blocker alone, offering a cue for discussion on the not yet well-defined interaction between the cardiac RAS and FGF23 signaling at the heart level [126]. The same study retrospectively evaluated 20 adolescents on maintenance hemodialysis, finding a protective effect on LVH with high-dose paricalcitol (15 mcg/week) [126]. 

Only two randomized controlled trials, PRIMO and OPERA, have assessed the protective effect of vitamin D treatment on LVH in CKD patients. Both trials, employing paricalcitol as VDRA, failed to demonstrate any effect on LVH regression or prevention [131,132]. 

The PRIMO study was a double-blind controlled trial that randomized 227 CKD patients with eGFR 15–60 mL/min, mild to moderate LVH, and preserved left ventricular ejection fraction to oral paricalcitol, 2 μg/d or placebo for 48 weeks. The trial failed to reach the primary endpoint that was a change in left ventricular mass index by cardiovascular magnetic resonance imaging. Low-dose paricalcitol and a small sample size have been addressed as major criticisms, as well as low PTH levels and hypercalcemia episodes in the treatment group, which would have limited any up-titration in paricalcitol dose [132]. 

The OPERA trial had the same study design and the same CKD severity population as PRIMO, except for a lower paricalcitol dose (1 μg/d). Despite the lower hospitalization rate and the effective control of PTH in the treatment group, paricalcitol did not influence LVH after an observation period of 52 weeks [131]. 

There are probably several reasons for such unsatisfactory results. Too low baseline paricalcitol doses, significant hypercalcemia episodes, and excessive PTH-lowering effects, requiring drug interruption, played a major role. Moreover, differently from the adolescent maintenance hemodialysis population studied by Czaya et al. [126], these patients were exposed to long-term comorbidity (mainly hypertension) that represents a major determinant of LVH. Further reasons for clinical trial failure should be sought in the complex interplay between calcitriol, circulating FGF23, FGFR4 signaling pathway, and VDR activation discussed above. 

In summary, VDRAs stimulate FGF23 production from osteocytes, determining a further increase in circulating FGF23 levels, which are already high in CKD. Extremely high FGF23 may counteract the protective effect of paricalcitol on LVH. Neither the PRIMO nor the OPERA trial assessed FGF23 levels, raising the suspicion that patients with higher FGF23 may benefit from higher VDRA doses. 

Besides VDRAs, a recent RCT addressed the effect of cholecalciferol on LVH (evaluated with cardiac MRI) in 49 stage 3–4 CKD patients with circulating vitamin D < 75 nmol/L with a 52-week follow-up. The trial, whose rationale resided in the activity of 1α-hydroxylase enzyme in non-renal tissues, including heart and blood vessels, failed to demonstrate any protective effect of native vitamin D on LVH. However, in contrast with the PRIMO and the OPERA trials, no effect on PTH or calcium levels was recorded [133]. Another possible explanation of such negative results may be the attenuated effect on RAS inhibition of both VDRAs and cholecalciferol since most patients (near 80%) of both trials were treated with RAS blockers and this therapy was an inclusion criterion in the trial by Banarjee et al. [133]. 

Table 1 summarizes the most relevant studies on the different vitamin D formulations used in clinical nephrology and their effects on PTH levels in CKD patients (non-dialysis-dependent and dialysis-dependent patients) and KTRs.

## 5. Conclusions 

Taken together, the currently available evidence suggests that in patients with CKD, nutritional vitamin D formulations (ergocalciferol, cholecalciferol, and calcifediol) are less effective in PTH suppression and management of SHPT compared to activated compounds, and this might mainly be due to the reduced activity of 1-α hydroxylase (CYP27B1) related to renal failure. However, in spite of the poor efficacy of nutritional vitamin D in SHPT control, it is associated with a reduced risk of hypercalcemia and/or hyperphosphatemia, or other adverse effects. Therefore, these points should be also considered by clinicians when establishing the first-line treatment, before starting a more aggressive treatment to correct poor vitamin D status in CKD progression, to lower cardiovascular risk, and to ameliorate kidney transplant outcomes.

## Figures and Tables

**Figure 1 nutrients-13-01453-f001:**
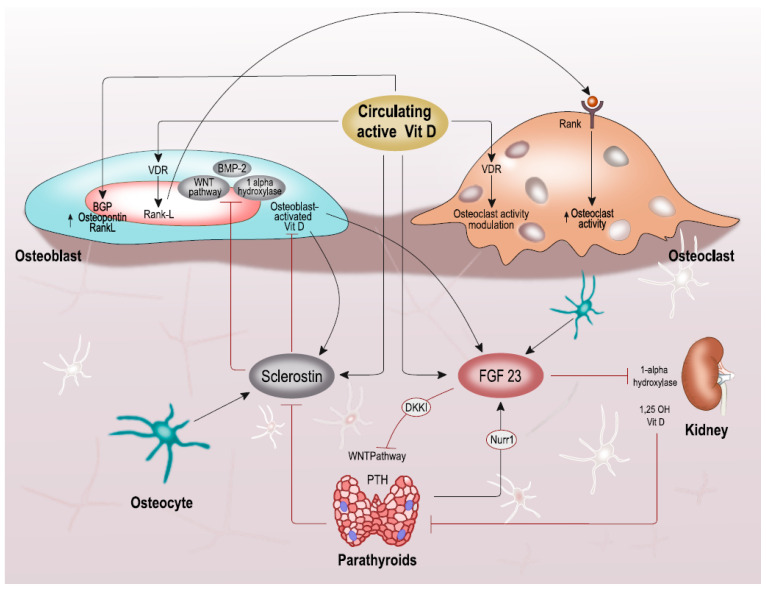
The complex interplay between active vitamin D (both circulating and activated by osteoblasts), osteocyte-produced FGF23 and sclerostin, parathyroid gland signaling, and effects on bone cells. Abbreviations: FGF23, fibroblast growth factor 23; BMP-2, bone morphogenic protein-2; Nurr-1, nuclear receptor-related protein-1; DKKI, dickkopf-related protein 1.

**Figure 2 nutrients-13-01453-f002:**
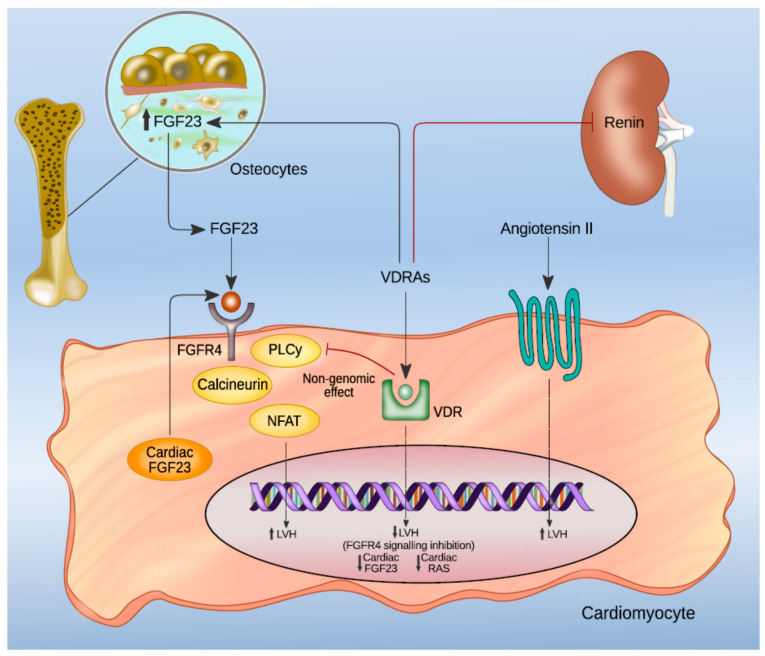
Interactions between active vitamin D and FGF23/FGFR4 signaling regarding LVH development at cardiomyocyte level. Active vitamin D counterbalances FGF23-induced LVH both by modifying gene expression and through a direct inhibitory effect on FGF23/FGFR4 transduction pathway. Abbreviations: FGF23, fibroblast growth factor 23; VDRAs, vitamin D receptor agonists; FGFR4, fibroblast growth factor receptor 4; PLCy, phospholipase Cy; NFAT, nuclear factor of activated T cells; LVH, left ventricular hypertrophy.

**Table 1 nutrients-13-01453-t001:** Vitamin D formulations used in clinical nephrology and most representative * studies on the effects of vitamin supplementation on PTH levels in CKD patients and KTRs.

Non-Dialysis-Dependent Patients										
Vitamin D Formulation	Type (N = Nutritional; A = Active)	Ref.	Design of the Study	N. of Patients	Dosage	Length of Therapy	25(OH)D	PTH	Ca	P
Ergocalciferol	D_2_, inactive prepro- hormone (N)	[48]	Retrospective	88	KDOQI guidelines	6 mo	↑	↓	~	~
		[49]	Open label RCT	68 (34 vs. 34)	Double vs. standard dose KDOQI guidelines	8 wk	↑	↓	~	~
Cholecalciferol	D_3_, inactive prepro- hormone (N)	[50]	Placebo-controlled RCT	46	50,000 IU/wk for 12 wk, then 50,000 IU every other wk for 40 wk	1 yr	↑	↓	~	~
		[53]	Double-blind RTC	95	8000 IU/d	12 wk	↑	~	~	~
25(OH)D (calcidiol, calcifediol)	D_3_, prehormone (N)	[54]	Secondary analysis of pooled data from 2 RCTs	429	30 μg daily oral dose of ERC	26 wk	↑	↓	~	~
Calcitriol		[60]	Systematic review of 16 studies	894	Various	Various	↑	↓	↑	↑
Paricalcitol	Vitamin D_2_ analog, VDRA (A)	[60]	Systematic review of 16 studies	894	Various	Various	↑	↓	↑	↑
Doxercalciferol	Vitamin D_2_ analog, VDRA (A)	[60]	Systematic review of 16 studies	894	Various	Various	↑	↓	↑	↑
22-oxacalcitriol	Vitamin D_3_ analog, VDRA (A)	[60]	Systematic review of 16 studies	894	Various	Various	↑	↓	↑	↑
Maxacalcitol	Vitamin D_3_ analog, VDRA (A)	[60]	Systematic review of 16 studies	894	Various	Various	↑	↓	↑	↑
**Dialysis-Dependent Patients**										
**Vitamin D Formulation**	**Type** **(N = Nutritional** **A = Active)**	**Ref.**	**Design of the Study**	**N. of Patients**	**Dosage**	**Length of Therapy**	**25(OH)D**	**PTH**	**Ca**	**P**
Ergocalciferol	D_2_, inactive prepro- hormone (N)	[63]	Placebo-controlled RCT	105	50,000 IU/wk vs. 50,000 IU/mo	12 wk	↑	~	~	~
Cholecalciferol	D_2_, inactive prepro- hormone (N)	[66]	Prospective	158	From 2700 IU thrice/wk to 50,000 IU/wk, based on 25(OH)D levels	1 year	↑	↓	↓	↓
25(OH)D (calcidiol, calcifediol)	D_3_, prehormone (N)	[68]	Systematic review of 60 studies	2773	Various	Various	↑	↓	↑	↑
Paricalcitol	Vitamin D_2_ analog, VDRA (A)	[68]	Systematic review of 60 studies	2773	Various	Various	↑	↓	↑	↑
Doxercalciferol	Vitamin D_2_ analog, VDRA (A)	[68]	Systematic review of 60 studies	2773	Various	Various	↑	↓	↑	↑
22-oxacalcitriol	Vitamin D_3_ analog, VDRA (A)	[68]	Systematic review of 60 studies	2773	Various	Various	↑	↓	↑	↑
Maxacalcitol	Vitamin D_3_ analog, VDRA (A)	[68]	Systematic review of 60 studies	2773	Various	Various	↑	↓	↑	↑
**Kidney Transplant Recipients**										
**Vitamin D Formulation**	**Type** **(N = Nutritional;** **A = Active)**	**Ref.**	**Design of the Study**	**N. of Patients**	**Dosage**	**Length of Therapy**	**25(OH)D**	**PTH**	**Ca**	**P**
Cholecalciferol	D_3_, inactive prepro- hormone (N)	[79]	Prospective controlled trial	90	25,000 IU/mo	1 year	↑	↓	~	↓
Cholecalciferol	D_3_, inactive prepro- hormone (N)	[80]	Prospective RCT	94	100,000 IU every 2 wk for 2 mo, then every other mo	1 year	↑	↓	↑ ~	↑ ~
25(OH)D (calcidiol, calcifediol)	D_3_, prehormone (N)	[81]	Double-blind prospective RCT	86 (45 calcitriol vs. 41 placebo)	Intermittent calcitriol (0.5 μg/48 h) in the first 3 mo, plus oral calcium (0.5 g/d) during 1 yr vs. calcium alone	1 year	↑	↓	~	~
Calcitriol	Active form of vitamin D_3_ (A)	[82]	Prospective RCT	86 (65 alfacalcidol + calcium vs. 46 no Treatment)	0.25 μg/d	6 mo	↑	↓	~	~
Alfacalcidol	Vitamin D_3_ analog, VDRA (A)	[83]	Prospective RCT	43	1 μg/d for 3 mo and then 2 µg/d (if tolerated)	6 mo	~	↓	~	~
Paricalcitol	Vitamin D_3_ analog, VDRA (A)	[84]	Prospective, open-label RCT	77 (37 calcitriol vs. 40 controls)	2 μg/d	44 wk	n.a.	↓	~	~
Doxercalciferol	Vitamin D_2_ analog, VDRA (A)	[84]	Prospective, open-label RCT	77 (37 calcitriol vs. 40 controls)	2 μg/d	44 wk	n.a.	↓	~	~

Abbreviations: CKD, chronic kidney disease; d, day(s); ERC, extended release calcifediol; HD, hemodialysis; KTRs, kidney transplant recipients; mo, month(s); n.a., not analyzed; PTH, intact parathyroid hormone; 25(OH)D, 25-hydroxyvitamin D; VDRA, vitamin D receptor activator; yr(s), year(s).

## Data Availability

Not applicable.

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
