# Peer review of "Vitamin D Effects on Bone Homeostasis and Cardiovascular System in Patients with Chronic Kidney Disease and Renal Transplant Recipients"

_nutrients, 2021, doi:10.3390/nu13051453_

Round 1

Reviewer 1 Report

The review is interesting, but needs some restructuring to make it more readable. The introduction is somewhat unclear as to making a point of the relevance and need for the review. There are a few factual mistakes (f.ex vitamin D2 is not synthesized in humans and it is unclear what is meant by 'the levels of circulating 25-OH-D only partly are dependent form the liver' as decribed in the first paragraph). Many studies are mentioned. That is ok, but the findings may to a larger extent be presented as tables. In general, I miss some interpretations of the scientific evidence and how these translate into advice for clinical practice. This may be summarized in a separate chapter of discussion. That said, the review is comprehensive, but may benefit on some restructuring and shortening of mechansitic explanations.

Author Response

The introduction is somewhat unclear as to making a point of the relevance and need for the review.

We have modified the introduction and better focused the aim of the study (first paragraph, line 80-85) in line with your suggestions.

Vitamin D2 is not synthesized in humans and it is unclear what is meant by the levels of circulating 25-OH-D only partly are dependent form the liver as decribed in the first paragraph.

The first was a typo; we modified (line 29-30) as “vitamin D2 is derived from the diet”. The second sentence (line 43-44) was linked to the possibility of circulating 25OH Vitamin D production by other tissues, like skin and testes.

The findings may to a larger extent be presented as tables.

This is a very valuable suggestion and we have reported in Table 1 the most important studies in CKD and KTRs patients.

In general, I miss some interpretations of the scientific evidence and how these translate into advice for clinical practice. This may be summarized in a separate chapter of discussion.

We have modified the title and the main focus of the paragraph 3, by adding the controversies in vitamin D supplementation, and we have explained in each sub-paragraph the possible clinical practical suggestions in CKD-MBD management.

Reviewer 2 Report

  1. The authors should emphasize better the controversial points existing at the moment in the scientific knowledge on vitamin D. The classical, broadly accepted information about vitamin D is certainly well presented and informative for a non-specialist reader. However, I believe the overall value of a new review on such a debated subject should focus on the innovative, and perhaps still debatable, notions. I would suggest a general reorganization of the text (and the use of new subsections where needed) to make the most from what it is, in most cases, already existing information.
  2. In sections aiming to tackle controversies around the vitamin D topic, the authors should consider presenting us the search string(s) which led them to select the quoted studies. Such data might maximize its relevance for the reader. If needed, they may consider adding new tables (or other forms useful for a synthetical presentation of such information) to the existing one, thus helping the reader select each subtopic’s main issues.
  3. Quotation of international and national guidelines, consensuses and position statements is undoubtedly relevant for the paper. However, while international documents keep a relative diversity in organizational sources, the national guidelines tend to limit to the Italian ones. Where applicable, a comparison with other national guidance papers from similar or different geographical areas may prove beneficial for the reader, which may have any other nationality as well as Italian.

Author Response

The authors should emphasize better the controversial points existing at the moment in the scientific knowledge on vitamin D. The classical, broadly accepted information about vitamin D is certainly well presented and informative for a non-specialist reader. However, I believe the overall value of a new review on such a debated subject should focus on the innovative, and perhaps still debatable, notions. I would suggest a general reorganization of the text (and the use of new subsections where needed) to make the most from what it is, in most cases, already existing information.

We have modified the title and partly the content of the paragraph number 3. We added the controversial conditions in vitamin D supplementation and VDRA therapy. Then, we have explained in each sub-paragraph the possible clinical practical suggestions in CKD-MBD management in CKD patients and KTRs.

In sections aiming to tackle controversies around the vitamin D topic, the authors should consider presenting us the search string(s) which led them to select the quoted studies.

Along with this suggestive remark, we have considered the most important studies that enrolled both non-dialysis-dependent and dialysis-dependent patients, and KTRs with significant population sample size. At the same time, we considered studies that analyse not only the effect on mineral metabolism but on cardiovascular and bone outcomes. We have reported in Table 1 the most relevant studies in CKD and KTRs patients.

Where applicable, a comparison with other national guidance papers from similar or different geographical areas may prove beneficial for the reader, which may have any other nationality as well as Italian.

We have cited both Italian, European (KDIGO) and American (KDOQI) guidelines. According to your suggestion, then we added Canadian Health Care guidelines in both CKD patients (295-299) and KTRs (526-528). At the same time, we reported that other experts (NICE, European Best Practice Guidelines) did not deviate from the indications provided by other authors. 

Round 2

Reviewer 2 Report

The authors thoroughly responded to all comments, allowing the overall quality of the paper to improve considerably.

I think the journal can consider the current form of this manuscript for publication.